# Determinants of Smoking Cessation Outcomes and Reasons for Relapse in Patients Admitted to a Smoking Cessation Outpatient Clinic in Turkey

**DOI:** 10.3390/ijerph21030310

**Published:** 2024-03-07

**Authors:** Tijen Acar, Claire Gallagher, Yasemin Gören, Bircan Erbas, Adem Özkara

**Affiliations:** 1Department of Family Medicine, School of Medicine, Kafkas University, 36000 Kars, Turkey; tijensengezer@hotmail.com; 2Centre of Epidemiology and Biostatistics, School of Population and Global Health, University of Melbourne, Parkville, VIC 3052, Australia; chgallagher@student.unimelb.edu.au; 3Department of Family Medicine, Bozüyük State Hospital, 11300 Bilecik, Turkey; yasemin.unsal1@saglik.gov.tr; 4Department of Public Health, School of Psychology & Public Health, La Trobe University, Bundoora, VIC 3086, Australia; 5Department of Family Medicine, Ankara Bilkent City Hospital, University of Health Sciences, 34668 İstanbul, Turkey

**Keywords:** smoking cessation, smoking relapse, smoking reuptake

## Abstract

The aim of this study was to identify the determinants of smoking cessation outcomes and reasons for relapse following smoking cessation treatment. Using a mixed-method design, 179 patients were recruited from the Smoking Cessation outpatient clinic of Ankara Numune Training and Research Hospital between May 2016 and May 2017. Quantitative data were collected via questionnaires or from patient files and qualitative data were obtained via 5 focus group interviews with 28 patients who relapsed to smoking following treatment. The success rate of the smoking cessation clinic at the end of one year was 26%. The number of applications to the clinic was significantly higher in the group who quit smoking. Treatment success was found to be higher in the group that applied behavioral recommendations. In focus group interviews with patients who relapsed, the most common causes were stressful events, especially workplace problems and serious health problems experienced by relatives. The presence of smokers in the immediate vicinity increased the risk of relapse. It was concluded that not stopping treatment before the recommended period, continuity in follow-up appointments, support of the environment, support of pharmacotherapy with cognitive behavioral therapy and improving patients’ coping skills were important.

## 1. Introduction

Tobacco smoking is a leading cause of global mortality and morbidity, responsible for more than 8.7 million deaths and nearly 230 million disability-adjusted life years (DALYs) in 2019 [1]. Despite worldwide efforts to reduce and prevent tobacco smoking, an estimated 1.14 billion people aged 15 years and older continue to smoke, with prevalence rates as high as 31.6% in Turkey (44.1% of men and 19.2% of women) [2,3].

Although the majority (70%) of people who smoke desire to quit, smoking cessation attempts often fail, with relapse rates as high as 97% in unaided quitters and 50% in those who attempt to quit with pharmacological treatments (nicotine replacement therapy (NRT), Bupropion and varenicline) [4,5,6]. Consequently, relapse to smoking remains a major challenge in reducing the burden of tobacco, and research suggests that achieving successful cessation takes, on average, more than 30 quit attempts [7].

Understanding factors influencing the effectiveness of interventions on long-term abstinence, including motivators to smoke and perceived barriers to successful cessation, is necessary to inform the design of effective smoking treatment and to guide practice in supporting long-term smoking abstinence. Therefore, this study employed a mixed-method design to (1) identify the primary factors associated with smoking cessation outcomes following smoking cessation treatment, and (2) understand motivators for smoking relapse and perceived barriers to successful cessation in patients who relapsed to smoking following cessation treatment.

## 2. Materials and Methods

### 2.1. Study Design

In this mixed-methods study, cross-sectional data were analyzed to describe and identify factors associated with smoking cessation, relapse, or continued smoking in recipients of smoking cessation treatment. Patients who relapsed were invited to attend focus group interviews to explore in-depth motivators of smoking reuptake. Ethics was approved by the Clinical Research Ethics Committee of Ankara Numune Training and Research Hospital (9 August 2018/E-18/2198).

### 2.2. Participant Selection

Participants were recruited from the Smoking Cessation Outpatient Clinic of Ankara Numune Training and Research Hospital between May 2016 and December 2017. Of the 236 patients eligible for recruitment, only 196 patients were reached, and 179 patients agreed to participate. We aimed to conduct focus group interviews with 30 patients who relapsed following treatment. A total of 93 patients were invited to focus group interviews, of which 28 patients agreed to participate.

### 2.3. Data Collection

#### 2.3.1. Quantitative Data

For the quantitative part of this mixed-methods study, the following demographic and health-related data were obtained by clinic physicians during the initial admission consultations with patients: age (year), gender (male, female), marital status (married, single or widowed), education level (primary, secondary, high school, university graduate), co-morbid diseases and alcohol use (yes, no). Psychiatric examinations were also performed by clinicians to assess the level of nicotine addiction, depression severity and anxiety severity. Specifically, the level of nicotine addiction was assessed using the Fagerstrom Test for Nicotine Dependence, which measures addiction on a scale ranging from 0 to 10. Higher scores on this scale indicate a greater degree of nicotine dependence. Nicotine addiction was analyzed both as a continuous variable and as a categorical variable, with the following classifications: very low dependence (0–2), low dependence (3–4), moderate dependence (5), high dependence (6–7), and very high dependence (8–10). The Beck Depression Inventory was used to measure depressive symptoms on a scale of 0–63. Depression severity was classified as minimal (0–13), mild (14–19), moderate (20–28), and severe (29–63). Similarly, the Beck Anxiety Inventory was used to measure anxiety symptoms on a scale of 0–63. Anxiety severity was classified as minimal (0–7), mild (8–15), moderate (16–25), and severe (26–63). In addition, according to the groups who continued smoking after the treatment, who relapsed, and who ceased smoking, the number of admissions, treatment method, treatment duration, and application status of behavioral changes were recorded from the face-to-face interviews held in the office during patient follow-ups using the 22-item author created data-collecting tool.

#### 2.3.2. Qualitative Data

All 93 patients who relapsed to smoking following cessation treatment were invited and 28 patients (18 women and 10 men) volunteered for focus group interviews. Five focus group interviews consisting of 4–7 participants were conducted over a 6-month period. Before starting the focus group discussions, the participants gave written consent. The interviews were conducted in a meeting room around a large table. The moderator asked 10 pre-prepared questions (Appendix A). The questions were mainly about the participants’ general thoughts on smoking behavior, the causes of relapse, their views on relapse prevention, their thoughts on quitting smoking, and their views on professional support. Every interview lasted 90 min. The moderator and co-moderator noted everything discussed in the session and conduct analyses were conducted.

### 2.4. Data Analysis

Data analysis was performed using IBM SPSS Statistics version 17.0 software (IBM Corporation, Armonk, NY, USA). Binary and nominal data were expressed as numbers (n) and percentages (%) and ordinal data were expressed as medians (min–max). Differences in ordinal variables were compared across groups (ceased smoking, relapsed to smoking and/or continued smoking) using the Mann–Whitney U test or the Kruskal–Wallis test, as appropriate. When the *p*-values from the Kruskal–Wallis test were statistically significant (*p* < 0.05), Conover’s multiple comparison test was used to identify which groups differed. Associations between binary and/or nominal variables were estimated using Pearson’s Chi-square test. However, to avoid issues related to small expected cell frequency and to improve accuracy, we used the continuity corrected Chi-square test if at least one cell had an expected frequency between 5 and 25, and the Fisher’s exact test if at least one cell had an expected frequency <5. In the event that at least 25% of cells had an expected frequency ≤5, the Likelihood Ratio test was used.

Variables hypothesized to predict smoking relapse or continued smoking were entered in univariate logistic regression models. Variables with a *p* < 0.10 were considered potential risk factors and were included in a multivariate binary and nominal regression model to determine the most important factor(s) in differentiating participants who ceased smoking from participants who continued smoking and/or relapsed to smoking. Odds ratios, 95% confidence intervals and Wald statistics are presented. A *p* value < 0.05 was considered statistically significant. However, for all possible multiple comparisons, the Bonferroni Correction was applied for controlling Type I errors.

## 3. Results

### 3.1. Characteristics of Patients

In our study, 22.3% (40) of the patients continued smoking after treatment, 52.0% (93) ceased smoking and then relapsed, and 25.7% (46) successfully ceased smoking. There were no significant differences in age, gender, marital status, education level, concomitant disease (excluding respiratory tract) and alcohol history between patients successful in smoking cessation and patients who were unsuccessful (continued smoking or relapsed after treatment) (Appendix A). However, the proportion of patients with respiratory tract disease was greatest in the group that successfully ceased smoking (*p* = 0.035).

Regarding patients’ smoking history, there were no significant differences in the age of smoking onset and the number of previous attempts to quit smoking between groups (Table 1). However, patients unsuccessful in smoking cessation consumed significantly more cigarettes per day prior to treatment (*p* < 0.001), were more likely to smoke at home or work (*p* = 0.042 and *p* = 0.026, respectively) and reported a greater degree of nicotine addiction (*p* < 0.001). No significant differences in the level of depression and anxiety before treatment were reported between groups (*p* = 0.088 and *p* = 0.337).

Across patients who ceased smoking, continued smoking or relapsed after treatment, we found significant differences with regard to the number of previous admissions to the smoking cessation clinic, the type and duration of treatment and compliance with suggested behavior change (Table 2). Specifically, patients who successfully ceased smoking had more previous admissions to the clinical than the continued or relapsed groups (*p* < 0.001). Patients who continued smoking were more likely to have received psychotherapy treatment only (*p* < 0.001) or Bupropion treatment (*p* = 0.037) than patients who successfully ceased smoking or relapsed. Most patients who continued smoking received treatment for a duration <4 weeks, whereas the majority of patients who relapsed received treatment for 4–7 weeks and most patients who ceased smoking received treatment for 12 weeks or more (*p* < 0.001). Additionally, patients who ceased smoking and relapsed were more likely to have always applied the recommended behavior change, whilst patients who continued smoking were more likely to have never applied the recommended behavior change.

### 3.2. Factors Associated with Unsuccessful Smoking Cessation

The results of multivariate logistic regression models indicate that the most important factors in predicting unsuccessful cessation following treatment are the number of previous admissions to the smoking cessation outpatient clinic and the degree of nicotine addiction (Table 3). Specifically, we found that the greater the number of previous admissions, the lower the likelihood of unsuccessful cessation (OR: 0.16, 95%CI: 0.06, 0.39), and the greater the level of nicotine addiction, the greater the likelihood of unsuccessful cessation (OR: 1.79, 95%CI: 1.15, 2.78).

In the group that continued smoking (Table 4), a greater number of previous admissions (OR: 0.13, 95% CI: 0.05, 0.36) and receiving varenicline treatment (OR 0.13, 95%CI: 0.02, 0.72) were associated with lower odds of unsuccessful cessation, whereas in the group that relapsed, a greater number of previous admissions (OR: 0.15, 95% CI: 0.06, 0.37) and a lower level of nicotine addiction (OR 1.95, 95%CI: 1.30, 2.95) were associated with lower odds of unsuccessful cessation. In this group, there was also moderate–weak evidence that having a respiratory disease reduced the likelihood of unsuccessful cessation (OR: 0.25, 95%CI: 0.06, 1.00, *p* = 0.05).

### 3.3. Duration of Cessation and Reasons for Smoking Relapse

Among patients who relapsed to smoking, 78% relapsed within the first 6 months (Appendix A). Of these, 56.7% relapsed in the first 3 months after treatment and 22% relapsed between the 4th and 6th month. The most common reasons for smoking reuptake were stress (37.6%), being in a smoking environment (28%) and sadness (22.6%) (Appendix A).

### 3.4. Differences in Perceptions of Smoking Cessation

After the treatment, the desire to quit smoking was significantly higher in patients who relapsed compared to patients who continued smoking (*p* = 0.006) and the relapsed group was less likely to think the treatment was useless (*p* = 0.002; Appendix A). No other differences in perceived obstacles to quitting were observed across groups.

There was a significant difference across groups in terms of the thought that quitting smoking is a problem of willpower (*p* = 0.008; Appendix A). The rate of those who thought that they had a willpower problem was significantly lower in patients who ceased smoking compared to patients who continued smoking or relapsed (*p* = 0.037 and *p* = 0.010).

Before treatment, there were no significant differences in the perceived difficulty of smoking cessation between those who ceased smoking and those who relapsed (*p* = 0.427; Appendix A). However, after treatment, the perceived difficulty of smoking cessation was significantly lower in those who ceased smoking (*p* < 0.001).

### 3.5. Factors Predicting Successful Cessation

The rate of those who thought health anxiety was a success factor was significantly higher in patients who successfully ceased smoking compared to patients who relapsed (*p* = 0.008; Appendix A). However, other factors such as medication support, individual motivation and social motivation did not significantly differ across groups (*p* > 0.05).

### 3.6. Treatment Side-Effects and Reported Changes Post-Treatment

There were no significant differences in the incidence of side effects across groups (Appendix A). However, with regard to negative changes, patients who ceased smoking were more likely to report weight gain than patients who relapsed (*p* = 0.045; Appendix A). No other differences in positive or negative changes were observed between groups.

### 3.7. Qualitative Results

Five focus group interviews were conducted with a total of 28 participants (18 women, 10 men). Focus group discussions started with the question of “Why do people smoke?” to understand how the participants perceived their smoking behavior. Participants answered the question as “due to stress”, “for spare time (habit)”, “to take a break, to rest”, “when they are happy/unhappy”, “it is a good friend”, “it is like a duty”, “a need” and “an enemy who defeated people”.

The answers of the participants who associated smoking with their mood can be exemplified as follows:

“I used to smoke especially in troubled times, but such an addiction has occurred that I also smoke when I am happy” (F, 63, fg2)

“People smoke when they are happy or unhappy; at most they smoke when they are sad” (F, 36, fg1)

Those who consider smoking to be a habit stated that they smoked more after meals, after tea/coffee, to take a break and to rest. They also mentioned that smoking was a necessity. However, the participants stated that the habit turned into a task over time. Sample answers have been presented below.

“Smoking is duty like eating” (F, 46, fg1)

“One gets used to smoking, habits are formed. For example, after meals, I had difficulties after quitting” (M, 52, fg2)

While most of the participants defined smoking as a “good friend”, only one participant defined smoking as “an enemy that beat me and I hate smoking” (F, 50, fg2). The answers of some of the participants who considered smoking to be a friend are as follows:

“Smoking is a good friend for me” (M, 40, fg1)

“I get discharged with smoking, I become more introverted especially during distress, I relieve my distress with my cigarette” (F, 63, fg2)

While most of the participants considered smoking to be a disease, four people defined it as a habit. When asked what kind of disease it is, only three participants answered “addiction”. Participants were also asked about the reasons for relapse and the moments when they experienced relapse. Most participants stated that they started smoking again during a stressful event. A participant described his thoughts as follows:

“Due to the negative words of my supervisor at work, I thought that only one would not do any harm. I forcibly took a cigarette from a friend and smoke. If I didn’t smoke, I would have hit someone. I also smoke one at home because of my husband’s debt. It was better than 40 friends while crying at that moment” (F, 40, fg4)

Participants stated that they smoked their first cigarette as a result of their self-confidence at times of stress. It was a common belief among participants that their first cigarette would be a one-off occurrence and that it would not become a habit. They stated that when they smoked their first cigarette, they had wanted to know how it would make them feel. Generally, the first cigarette smoked was offered to them by smokers in the environment at that moment. Only one participant stated that he bought his first cigarette himself.

Other conditions that caused relapse were stress at work, family problems, loss of relatives, divorce, the presence of smokers in the home or in the immediate vicinity, inability to cope with problems, loss of motivation and difficulties (gaining weight, unhappiness, malaise, etc.) in the smoking cessation process.

Most of the participants agreed that they would not start smoking again if there was no one around whom they could want to smoke. In addition, being able to say no when offered cigarettes, finding occupations to keep themselves busy, taking up new hobbies, staying away from the smoking environment and being determined not to smoke again were the solutions offered by participants. Participants who said that they were not informed about relapses during treatment thought that they would have acted differently if they had been informed. Sample comments are as follows:

“It is not possible at all not to smoke when smokers are around all the time” (M, 35, fg5)

“Professional support is critical, especially during difficult times” (F, 44, fg3)

Of the 28 participants, 3 had no intention to quit smoking before the interview. Although the participants shared a common opinion on the harms of smoking and its economic negative effects and stated that smoking is a disease, the number of participants who stated that they did not have sufficient will to quit smoking was also high.

When the participants were asked for their opinions about obtaining professional support, the opinion that smoking cessation was related to personal will was quite high, but there were also those who thought that they would have been successful if they could comply with the suggested behavioral changes. In addition to those who thought that it would be beneficial to comply with behavior change, there were also those who thought that accessing someone who could obtain professional support in case of craving for cigarettes would prevent relapse. However, it was observed that patients who quit and started again were hesitant to apply to the smoking cessation outpatient clinic again due to prejudices, such as spending unnecessary time and encountering a negative reaction from the experts.

## 4. Discussion

In our study, relapses, which constitute a major obstacle in smoking cessation treatment, and factors that may be related to post-treatment failure, were evaluated. In the literature, the success rate of smoking cessation outpatient clinics varied between 21 and 45% and a similar rate (26%) was achieved in our study [8].

We found that socio-demographic characteristics such as age, gender, marital status and education level did not affect the success of smoking cessation. In some studies, it was shown that elderly patients had higher smoking cessation success than younger patients [8,9]. Although 57.5% of our participants were aged 40 and over, rates of smoking cessation did not differ across age groups.

Consistent with the literature, stress was an important reason for smoking by the participants in our study [10,11,12]. It was also stated that smoking was a tool for spending time, resting and taking a mini break. Studies in the literature similarly reported that smoking was used to take a break and rest [13]. Further to this, most of our participants defined smoking as a “good friend”. In the study of Hendricks et al., the participants stated that smoking was a friend that relieved them [14]. Only three of our participants used the term “addiction” for cigarettes. These findings suggest that incorporating stress management and coping strategies into cessation treatment may reduce the likelihood of smoking relapse. This recommendation is supported by promising results from a systematic review [15], which found that mindfulness-based interventions had positive effects on cigarette cravings, smoking cessation and relapse prevention. However, further research is needed to determine the most feasible and effective interventions for stress management among smokers.

Current evidence on the effects of alcohol on smoking cessation and relapse is mixed. Whilst our study found no difference in rates of relapse between alcohol consumers and abstainers, an earlier Turkish study (n = 550) found that alcohol consumption increased the likelihood of relapse [16]. Other studies also report conflicting findings, with some suggesting alcohol consumption as a risk factor for relapse [17,18] and others finding no evidence of such a relationship [9,19].

We concluded that the success of smoking cessation decreased significantly as the number of cigarettes consumed per day increased. Our result is similar to other studies in the literature [9,20,21]. Consistent with the literature, we found that each step increase in the degree of Fagerström addiction statistically significantly increased the probability of treatment failure [21,22,23,24,25].

Although depression and anxiety levels were higher in the group who did not quit smoking after the treatment or who started smoking again after the treatment, there was no statistically significant difference between the groups. Similarly, in the study of Morozova et al. from 2015, no negative effect of depression on smoking cessation was found [26]. In another study, it was stated that the negative mood of individuals with depression was associated with increased relapse rates [27]. Yaşar et al. (2012) found that those with high depression and anxiety levels had lower success in quitting smoking [24]. Kaya et al. (2003) concluded that smoking cessation success rates were lower in participants with high depression and anxiety levels [28].

There are studies that show successful cessation increases with the number of quit attempts. However, in our study, there was no difference between the number of attempts to quit smoking and smoking cessation. This was also found by Kaya et al. and Chatkin et al. [23,28]. Instead, we found that as the number of previous admissions to our smoking cessation outpatient clinic increased, the probability of success increased. This is unsurprising, given that unaided quit attempts are considered the least effective method to successfully cease smoking. Rather, our findings highlight that smoking cessation is more successful when aided by treatment. Further to this, we found that the rate of relapse was reduced when treatment was not interrupted (not terminated prematurely). Yaşar et al. found that smoking cessation rates increased significantly in those who received pharmacological treatment for a sufficient period of time [24]. Tural Önür et al. concluded that the success of quitting is higher in patients who came to follow-up appointments [29].

The rate of those who continued smoking was significantly lower in the group receiving varenicline treatment and significantly higher in the group receiving Bupropion treatment. However, as very few patients were started on NRT and combined therapy, our analysis may have lacked the power to detect an effect in these treatment groups. In the Cochrane meta-analysis by Cahill et al. (2013), NRT and Bupropion were both superior to placebo with equal efficacy, however, varenicline was superior to single forms of NRT and Bupropion [30]. In a Cochrane review conducted in 2012, the number of people who quit smoking with varenicline was found to be higher than with NRT and Bupropion [31]. In the Eagles study, varenicline provided a significantly higher quit rate compared to Bupropion, NRT and placebo [32].

Relapses were observed most frequently between 1 and 3 months (30%). In the study of Koçak et al., relapse rates were found to be 48.6% in the first month and 37% in the sixth month [16]. In our study, the first 6 months were the months with the highest risk of relapse and at the end of 6 months, the rate of relapse decreased. Similar to our results, in another study, it was observed that the risk of relapse gradually decreased at the end of 1 and 2 years [33].

The presence of smokers in the immediate vicinity and a stressful event were identified as the main causes of relapse. In a qualitative study by Thompson et al., work stress was stated as a cause of relapse by the participants [13]. There are also findings in different studies emphasizing the importance of stress in relapse [11,13,14,34,35,36,37]. In the literature, there are many studies showing that smoking cessation success decreases in the presence of a smoker at home or work [9,10,16,24,38].

The importance of social support in the smoking cessation process was stated in the focus group interviews. In the study of Chandola et al., it was concluded that high social support increased the success of quitting [22]. In addition, it was determined the presence of smokers in the immediate vicinity and being offered cigarettes was an important cause of relapse [22,35,39].

The common idea of the participants who had relapsed was that they would no longer continue to smoke again as they had quit smoking. In the study of Turner et al., participants thought that smoking only one cigarette would not cause relapse [40]. Another reason for relapse was that automated behaviors and daily smoking rituals created a desire to smoke. In the study of Tanielu et al., it was concluded that automated behaviors increase the risk of relapse after smoking cessation [10]. Accordingly, the importance of cognitive behavioral therapy in quitting smoking was found to be high in our study, similar to previous studies [41,42].

We determined that those who could apply the behavioral change recommendations were able to quit smoking and achieve success in the treatment. In a meta-analysis study, pharmacotherapy supported by cognitive behavioral therapy was shown to increase smoking cessation rates [41].

Similar to the literature, we concluded that weight gain after smoking cessation negatively affected the success of treatment [14,36]. Similar to the studies of Ossler & Presscot and Patten et al., the fact that people could engage themselves and find new motivations was presented as a suggestion that can prevent relapse [11,20].

It was observed that most of the participants wanted to quit smoking again, but they had negative thoughts about quitting smoking. Hopelessness about not being able to quit was a key conclusion in the study by Thompson et al. [13]. Other factors that the participants thought were effective in their success were individual motivation and drug support. Most of the participants stated that professional support to quit smoking was important and they were more motivated when support was received. In the study by Thompson et al., the participants stated that it was necessary to obtain professional help [13]. In the study by Zhu et al., it was shown that receiving professional support increased the success of quitting and reduced relapse [43].

A major strength of this study is the mixed-method design, which combined quantitative and qualitative data to allow deeper insights into the determinants of smoking relapse. However, the application of mixed-methods designs is not without limitations, including challenges in the interpretation of conflicting results from the differing methods. We also determined smoking cessation based on patient reports without verifying their smoking status through carbon monoxide breath testing. Furthermore, our sample size was relatively small (n = 179) and patients were sampled from a single clinic, limiting the generalizability of findings. That said, our findings were mostly consistent with the results of studies published in other settings.

## 5. Conclusions

In conclusion, we determined that it is very important for family physicians to have sufficient information about people who are in the smoking cessation process and to know them closely. Accordingly, it is thought that being able to follow the patients closely, being in touch with their relatives and reminding them about how to support them are necessary for effective support in increasing the success in the smoking cessation process and preventing relapse. It is also considered important to inform the family physicians in smoking cessation outpatient clinics that they also receive smoking cessation treatment, as it will provide an advantage for effective support in relapse prevention.

## Figures and Tables

**Table 1 ijerph-21-00310-t001:** Smoking history and clinical characteristics of patients who were successful in smoking cessation compared to patients who were unsuccessful (continued smoking or relapsed after treatment).

	Total	Successful Cessation	Unsuccessful Cessation	*p*-Value
Age of first cigarette start				0.656 †
<12 years	7 (%4.1)	0 (%0.0)	7 (%5.6)	
12–17 years	83 (%48.5)	23 (%51.1)	60 (%47.6)	
18–25 years	75 (%43.9)	21 (%46.7)	54 (%42.9)	
>25 years	6 (%3.5)	1 (%2.2)	5 (%3.9)	
Daily cigarette smoking				**<0.001** †
1–5 cigarettes	4 (%2.3)	2 (%4.4)	2 (%1.5)	
6–10 cigarettes	25 (%14.1)	12 (%26.7)	13 (%9.8)	
11–20 cigarettes	82 (%46.3)	22 (%48.9)	60 (%45.5)	
21–30 cigarettes	41 (%23.2)	5 (%11.1)	36 (%27.3)	
>30 cigarettes	25 (%14.1)	4 (%8.9)	21 (%15.9)	
Number of quit attempts				0.335 †
None	34 (%19.6)	10 (%22.2)	24 (%18.7)	
1 time	75 (%43.4)	14 (%31.1)	61 (%47.7)	
2 times	29 (%16.8)	9 (%20.0)	20 (%15.6)	
3 times	19 (%11.0)	6 (%13.3)	13 (%10.2)	
4 and above	16 (%9.2)	6 (%13.3)	10 (%7.8)	
Cigarette smoking at home				**0.042** ‡
Yes	63 (%35.2)	10 (%21.7)	53 (%39.8)	
No	116 (%64.8)	36 (%78.3)	80 (%60.2)	
Smoking in the workplace				**0.026** ‡
Yes	128 (%74.4)	26 (%60.5)	102 (%79.1)	
No	44 (%25.6)	17 (%39.5)	27 (%20.9)	
Nicotine addiction score				**<0.001**
Very low	17 (%9.8)	10 (%22.7)	7 (%5.4)	
Low	38 (%22.0)	14 (%31.8)	24 (%18.6)	
Moderate	25 (%14.4)	7 (%15.9)	18 (%14.0)	
High	34 (%19.7)	7 (%15.9)	27 (%20.9)	
Very high	59 (%34.1)	6 (%13.7)	53 (%41.1)	
Beck depression level				0.088
No	73 (%50.3)	23 (%60.5)	50 (%46.7)	
Light	43 (%29.7)	11 (%29.0)	32 (%29.9)	
Moderate	25 (%17.2)	3 (%7.9)	22 (%20.6)	
Severe	4 (%2.8)	1 (%2.6)	3 (%2.8)	
Beck anxiety level				0.337
No	70 (%47.9)	22 (%56.4)	48 (%44.9)	
Light	44 (%30.1)	9 (%23.1)	35 (%32.7)	
Moderate	22 (%15.1)	5 (%12.8)	17 (%15.9)	
Severe	10 (%6.9)	3 (%7.7)	7 (%6.5)	

† Mann–Whitney U test, ‡ Chi-square test with continuity correction. Bold values indicate significant results.

**Table 2 ijerph-21-00310-t002:** Number of applications, treatment method, duration of treatment and behavioral changes of participants who ceased smoking compared to participants who continued smoking or relapsed.

	Continued Smoking	Relapsed	Ceased Smoking	*p*-Value
**Number of admissions ***				**<0.001** †
1 time	24 (%60.0) ^a^	58 (%62.4) ^b^	9 (%19.6)	
2–5 times	16 (%40.0)	32 (%34.4)	29 (%63.0) ^a,b^	
6 and above	0 (%0.0)	3 (%3.2)	8 (%17.4)	
**Treatment method**				
psychotherapy only	7 (%17.5) ^a,c^	3 (%3.2) ^c^	0 (%0.0) ^a^	**<0.001** ‡
varenicline	13 (%32.5) ^a,c^	60 (%64.5) ^c^	31 (%67.4) ^a^	**<0.001** ¶
bupropion	18 (%45.0) ^a,c^	24 (%25.8) ^c^	10 (%21.7) ^a^	**0.037** ¶
tape gum	1 (%2.5)	3 (%3.2)	4 (%8.7)	0.313 ‡
combined therapy	1 (%2.5)	3 (%3.2)	1 (%2.2)	0.930 ‡
**Duration of Treatment**				**<0.001** †
<4 weeks	26 (%74.3) ^a,c^	28 (%30.4)	9 (%19.6)	
4–7 weeks	5 (%14.3)	31 (%33.7) ^b,c^	7 (%15.2)	
8–11 weeks	3 (%8.6)	19 (%20.7)	11 (%23.9) ^a,b^	
12 weeks and above	1 (%2.8)	14 (%15.2)	19 (%41.3)	
**Suggested behavior change**				**<0.001** ‡
Always done	9 (%23.1) ^a,c^	48 (%51.6) ^c^	27 (%60.0) ^a^	
Sometimes done	23 (%59.0)	43 (%46.2)	17 (%37.8)	
Never done	7 (%17.9) ^a,c^	2 (%2.2) ^c^	1 (%2.2) ^a^	

* Smoking cessation outpatient clinic, † Kruskal–Wallis test, ‡ Likelihood Ratio test, ¶ Pearson’s Chi-square test. ^a^ The difference between the continued smoking group and the ceased smoking group is statistically significant (*p* < 0.05). ^b^ The difference between the ceased smoking and relapsed group is statistically significant (*p* < 0.001). ^c^ The difference between the continued smoking and relapsed group is statistically significant (*p* < 0.05). Bold values indicates significant results.

**Table 3 ijerph-21-00310-t003:** Multivariate logistic regression analysis to examine the combined effects of all possible factors associated with unsuccessful smoking cessation (continued smoking or relapsed).

	Odds Ratio	95% Confidence Interval	*p*-Value
Respiratory disease	0.428	0.123–1.494	0.183
Cigarettes per day	1.314	0.703–2.457	0.392
Smoking in the house	1.647	0.596–4.555	0.336
Smoking in the workplace	1.234	0.399–3.819	0.716
Nicotine addiction score	1.786	1.148–2.776	**0.010**
Beck depression level	0.917	0.481–1.748	0.792
Number of applications to outpatient clinic	0.159	0.064–0.391	**<0.001**

Bold values indicates significant results.

**Table 4 ijerph-21-00310-t004:** Multivariable ordinal logistic regression analysis exploring factors associated with continued smoking and relapse to smoking following cessation treatment.

	Odds Ratio	95% Confidence Interval	*p*-Value
Continued Smoking			
Education	0.60	0.26–1.37	0.227
Daily cigarette consumption	1.95	0.94–4.07	0.075
Fagerström addiction score	1.50	0.91–2.47	0.110
Number of admissions to smoking cessation clinic	0.13	0.05–0.36	**<0.001**
Smoking in the house	1.73	0.53–5.63	0.361
Smoking in the workplace	0.68	0.19–2.44	0.554
Taking varenicline therapy	0.13	0.02–0.72	**0.020**
Getting Bupropion Treatment	0.46	0.08–2.75	0.395
Respiratory disease	0.22	0.04–1.16	0.074
Relapsed to Smoking			
Education	0.93	0.45–1.91	0.840
Daily cigarette consumption	1.02	0.55–1.89	0.953
Fagerström addiction score	1.95	1.30–2.95	**<0.001**
Number of admissions to smoking cessation clinic	0.15	0.06–0.37	**<0.001**
Smoking in the house	2.42	0.90–6.51	0.081
Smoking in the workplace	1.23	0.41–3.64	0.711
Taking varenicline therapy	0.50	0.10–2.43	0.391
Getting Bupropion Treatment	0.54	0.10–2.94	0.473
Respiratory disease	0.25	0.06–1.00	0.050

Reference category is the ceased smoking group. Bold values indicates significant results.

## Data Availability

Data are not available due to ethical restrictions.

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
