# Peer review of "Determinants of Smoking Cessation Outcomes and Reasons for Relapse in Patients Admitted to a Smoking Cessation Outpatient Clinic in Turkey"

_ijerph, 2024, doi:10.3390/ijerph21030310_

Round 1

Reviewer 1 Report

Comments and Suggestions for Authors

From the title of the paper it can be noted that the topic discussed in the article is socially relevant in the line of research on the maintenance and decline of tobacco consumption, and on the other hand, due to the consequences of tobacco consumption. The role is relevant, because based on the results, specific strategies for the factors associated with smoking could be included in smoking cessation treatments with the purpose of maintaining long-term abstinence. However, it is important to review the following:

-              Review the consistency of the objective set in the research, in the summary the objective is indicated: “The objective of this study was to identify the determinants of smoking relapse after smoking cessation treatment”; and at the end of the introduction it is noted: “this study used a mixed methods approach to describe the general characteristics of patients admitted to the Smoking Cessation Outpatient Clinic in one year; determine the factors that increase or decrease the probability of successfully quitting smoking; and determine the deep motivators for smoking and perceived barriers to successful smoking cessation in patients who relapsed into smoking.”

-              The results of the statistical analyzes must be reported including the values ​​of the statistical tests used, not only the probability. Likewise, the necessary data must be reviewed and included to report the results of the multivariate logistic regression analysis carried out.

-              Likewise, in the report of the results of the focus groups, emphasis is placed on the reasons for relapse and the moments in which they experience a fall, which is consistent with the title but not with the objective set at the end of the intervention.

-       The same thing happens with the discussion, two aspects are recovered, the factors associated with relapse and the factors associated with the success of smoking cessation, which is why the objective of the paper must be reviewed.

Author Response

  1. Review the consistency of the objective set in the research, in the summary the objective is indicated: “The objective of this study was to identify the determinants of smoking relapse after smoking cessation treatment”; and at the end of the introduction it is noted: “this study used a mixed methods approach to describe the general characteristics of patients admitted to the Smoking Cessation Outpatient Clinic in one year; determine the factors that increase or decrease the probability of successfully quitting smoking; and determine the deep motivators for smoking and perceived barriers to successful smoking cessation in patients who relapsed into smoking.”

We agree there are inconsistencies in our statements of the study aim, thank you for alerting us to this oversight. We have now revised the study aim to be consistent with our research objective. We have also revised the title accordingly.

New Aim: This study employed a mixed-method design to: (1) identify the primary factors associated with smoking cessation outcomes following smoking cessation treatment, and (2) understand motivators for smoking relapse and perceived barriers to successful cessation in patients who relapsed to smoking following cessation treatment.

New Title: Determinants of Smoking Cessation Outcomes and Reasons for Relapse in Patients Admitted to a Smoking Cessation Outpatients Clinic in Turkey.

  1. The results of the statistical analyzes must be reported including the values ​​of the statistical tests used, not only the probability. Likewise, the necessary data must be reviewed and included to report the results of the multivariate logistic regression analysis carried out.

The presentation of test statistics is not a reporting requirement of IJERPH, and we have opted to omit this information to ensure a clear and concise summary of results so not to exceed word count restrictions. We believe that our presentation of summary statistics (descriptive results) or measures of association (inferential results), along with estimates of precision and p-values is sufficient and provides readers with the information necessary to interpret our results.

  1. Likewise, in the report of the results of the focus groups, emphasis is placed on the reasons for relapse and the moments in which they experience a fall, which is consistent with the title but not with the objective set at the end of the intervention.

Please see our response to Comment 1

  1. The same thing happens with the discussion, two aspects are recovered, the factors associated with relapse and the factors associated with the success of smoking cessation, which is why the objective of the paper must be reviewed.

Please see our response to Comment 1

Reviewer 2 Report

Comments and Suggestions for Authors

1. This efforts is overall commendable. Many patients enroll in smoking cessation clinics but rate if relapse is consistently seen to be high, and it needs to be understood why patients relapse. Authors have tried to understand from patients themselves, and as expected, stress emerges as a major reason. Accordingly, it would be reasonable to include stress reduction strategies in smoking cessation programs.  Suggestion to authors-cite some studies and/or make a comment on this. 

2. While this study is specific to Turkey, results seem universally applicable. Suggest compare with similar studies worldwide and give references.

3. Authors said rate if success increased as number of patients in smoking cessation clinic increased- this is not explained. Please add an explanation why should this be so.

Comments on the Quality of English Language

Acceptable 

Author Response

  1. This effort is overall commendable. Many patients enrol in smoking cessation clinics but rate if relapse is consistently seen to be high, and it needs to be understood why patients relapse. Authors have tried to understand from patients themselves, and as expected, stress emerges as a major reason. Accordingly, it would be reasonable to include stress reduction strategies in smoking cessation programs. Suggestion to authors-cite some studies and/or make a comment on this. 

We agree, stress is an important risk factor for relapse in our study. Whilst we already recommend stress-management and coping strategies as an intervention to reduce the likelihood of relapse, we have now included evidence from a systematic review exploring the effectiveness of mindfulness-based interventions on smoking cessation to support our recommendation.

OLD: Consistent with the literature, stress was an important reason for smoking by the participants in our study (12-14). It was also stated smoking was a tool for spending time, resting and taking a mini break. Studies in the literature similarly reported that smoking was used to take a break and rest (15). Further to this, most of our participants defined smoking as a “good friend”. In the study of Hendricks et al., the participants stated that smoking was a friend that relieved them (16). Only three of our participants used the term “addiction” for cigarettes. These findings suggest that incorporating stress management and coping strategies into cessation treatment may reduce the like-lihood of smoking relapse.

NEW: Consistent with the literature, stress was an important reason for smoking by the participants in our study (12-14). It was also stated smoking was a tool for spending time, resting and taking a mini break. Studies in the literature similarly reported that smoking was used to take a break and rest (15). Further to this, most of our participants defined smoking as a “good friend”. In the study of Hendricks et al., the participants stated that smoking was a friend that relieved them (16). Only three of our participants used the term “addiction” for cigarettes. These findings suggest that incorporating stress management and coping strategies into cessation treatment may reduce the likelihood of smoking relapse. This recommendation is supported by promising results from a systematic review (de Souza et al., 2015) which found mindfulness-based interventions had positive effects on cigarette cravings, smoking cessation, and relapse prevention. However, further research is needed to determine the most feasible and effective interventions for stress management among smokers.

  1. While this study is specific to Turkey, results seem universally applicable. Suggest compare with similar studies worldwide and give references.

Throughout our discussion, we have compared our findings with the results of other studies. Whilst our results are mostly in agreement, we are cautious not to over-generalise our findings, as our sample size was relatively small and our source population was only one smoking cessation clinic in Turkey. However, we have added to our discussion of limitations to highlight that similar findings are likely to be observed in other settings.

OLD: Furthermore, our sample size was relatively small (n=179) and patients were sampled from a single clinic, limiting the generalizability of findings.

NEW: Furthermore, our sample size was relatively small (n=179) and patients were sampled from a single clinic, limiting the generalizability of findings. That said, our findings were mostly consistent with the results of studies published in other settings.

  1. Authors said rate of success increased as number of patients in smoking cessation clinic increased- this is not explained. Please add an explanation why should this be so.

We originally wrote “as the number of admissions to our smoking cessation outpatient clinic increased, the probability of success increased.” What we meant by this is that as the number of previous admissions to our smoking cessation clinic increased, the more likely a patient was to successful cease smoking. As the language we used in the manuscript was not clear, we have revised our discussion to ensure people are aware we are referring to the number of previous admissions of a given patient and not the number of admissions to the clinic at a given time.

Furthermore, we talk about both the number of quit attempts and the number of previous admissions. Previously, these were discussed in separate paragraphs, however, to avoid confusion, we have decided to combine our discussion into one paragraph, so that reader are aware of the distinction between these variables.

OLD: There are studies that show successful cessation increase with the number of quitting attempts. However, in our study there was no difference between the number of attempts to quit smoking and smoking cessation. This was also found by Kaya et al. and Chatkin et al (17,18).

We found that as the number of admissions to our smoking cessation outpatient clinic increased, the probability of success increased. We have also found that the rate of relapse was reduced when treatment was not interrupted (not terminated prematurely). Yaşar et al. found that smoking cessation rates increased significantly in those who re-ceived pharmacological treatment for a sufficient period of time (26). Tural Önür et al. concluded that the success of quitting is higher in patients who came to follow-up ap-pointments (30).

NEW: There are studies that show successful cessation increases with the number of quitting attempts. However, in our study there was no difference between the number of attempts to quit smoking and smoking cessation. This was also found by Kaya et al. and Chatkin et al (17,18). Instead, we found that as the number of previous admissions to our smoking cessation outpatient clinic increased, the probability of success increased. This is unsurprising, given that unaided quit attempts are considered the least effective method to successfully cease smoking. Rather, our findings highlight that smoking cessation is more successful when aided by treatment. Further to this, we found that the rate of relapse was reduced when treatment was not interrupted (not terminated prematurely). Yaşar et al. found that smoking cessation rates increased significantly in those who received pharmacological treatment for a sufficient period of time (26). Tural Önür et al. concluded that the success of quitting is higher in patients who came to follow-up appointments (30).

Reviewer 3 Report

Comments and Suggestions for Authors

The study's goal is to investigate the factors that influence the quitting smoking success of patients who attend the smoking cessation outpatient clinic, as well as the rate of quitting smoking at this clinic. The quantitative data were collected using questionnaires or patient files, whereas the qualitative data were gained through five focus group interviews with 28 patients who reverted to smoking after 21 treatments. Finally, they concluded that it is critical for family physicians to know and understand persons who are in the process of quitting smoking. The study has so many limitations.

1) Sample size is insufficient.

2) Did you collect data on respiratory function tests, electrocardiography, biochemicals, and hematology?

3) How were subjects evaluated for depression or psychological issues?

4) Did the subjects have individual interviews?

5) In Data Analysis - Lines 104-106, were cells showed to have a frequency of 5-25? What does it imply?

Author Response

  1. Sample size is insufficient.

We recognize that our sample size is relatively small, however, we believe it is sufficient when compared to the size of the source population. Only 236 patients applied to the Smoking Cessation Outpatient Clinic of Ankara Numune Training and Research Hospital, University of Health Sciences, between May 2016 and May 2017. Whilst we were able to reach 196 patients, a further 17 patients decline to participate, reducing the analytic sample to n=179. Although small, our study sampled 76% of the source population, and whilst we believe this to be sufficient, we agree it is a limitation worth noting. Therefore, we have revised our discussion to include our small sample size as a possible limitation that may prevent generalizability.

OLD: However, the application of mixed-methods designs is not without limitations, including challenges in the interpretation of conflicting results from the differing methods. In addition, patients were sampled from a single clinic, limiting the generalizability of findings.

NEW: However, the application of mixed-methods designs is not without limitations, including challenges in the interpretation of conflicting results from the differing methods. We also determined smoking cessation based on patient report without verifying their smoking status through carbon monoxide breath testing. Furthermore, our sample size was relatively small (n=179) and patients were sampled from a single clinic, limiting the generalizability of findings.

Similarly, with regards to the focus groups, we planned to interview 30 participants but only 28 volunteered. Despite this, we conducted a total of five focus group interviews, and evidence (Guest, Namey & McKenna, 2017) suggests that three focus groups is sufficient to identify the more prevalent themes within a sample and that three to six focus groups is sufficient to discover 90% of all themes.

Guest, G., Namey, E., & McKenna, K. (2017). How Many Focus Groups Are Enough? Building an Evidence Base for Nonprobability Sample Sizes. Field Methods. 29(1), 3–22.

  1. Did you collect data on respiratory function tests, electrocardiography, biochemicals, and hematology?

Such data was not collected as it is not routine practice at the clinic to perform laboratory tests such as pulmonary function tests, electrocardiography, biochemicals and hematology. Rather, it is at the physician’s discretion to order tests if deemed necessary. We acknowledge that there are limitations in relying on patient report of smoking cessation without biological confirmation, and so we have revised our discussion of limitations accordingly.

NEW: However, the application of mixed-methods designs is not without limitations, including challenges in the interpretation of conflicting results from the differing methods. We also determined smoking cessation based on patient report without verifying their smoking status through carbon monoxide breath testing. Furthermore, our sample size was relatively small (n=179) and patients were sampled from a single clinic, limiting the generalizability of findings.

  1. How were subjects evaluated for depression or psychological issues?

Whilst we reported that depression and anxiety were assessed using the Beck Depression Inventory and Beck Anxiety Inventory, we agree that the detail we provided was not sufficient. We have now extended our methods to include details regarding the measurement, scoring and classification of nicotine dependence, depression, and anxiety.

OLD: For the quantitative part of this mixed-methods study, the following demographic and health-related data were obtained from patient files created during their initial admission: age (year), gender (male, female), marital status (married, single or widowed), education level (primary, secondary, high school, university graduate), co-morbid diseases, alcohol use (yes, no), Fagerström Nicotine Addiction Level, Beck Depression Scale and Beck Anxiety Scale.

NEW: For the quantitative part of this mixed-methods study, the following demographic and health-related data were obtained by clinic physicians during the initial admission consultations with patients: age (year), gender (male, female), marital status (married, single or wid-owed), education level (primary, secondary, high school, university graduate), co-morbid diseases and alcohol use (yes, no). Psychiatric examinations were also performed by clinicians to assess level of nicotine addiction, depression severity and anxiety severity. Specifically, level of nicotine addiction was assessed using the Fagerstrom Test for Nicotine Dependence which measures addiction on a scale ranging from 0 to 10. Higher scores on this scale indicate a greater degree of nicotine dependence. Nicotine addiction was analyzed both as a continuous variable and as a categorical variable, with the following classifications: very low dependence (0-2), low dependence (3-4), moderate dependence (5), high dependence (6-7), and very high dependence (8-10). The Beck Depression Inventory was used to measure depressive symptoms on a scale of 0-63. Depression severity was classified as minimal (0-13), mild (14-19), moderate (20-28), and severe (29-63). Similarly, the Beck Anxiety Inventory was used to measure anxiety symptoms on a scale of 0-63. Anxiety severity was classified as minimal (0–7), mild (8–15), moderate (16–25), and severe (26–63).

  1. Did the subjects have individual interviews?

Yes, when patients were admitted to the clinic, an individual interview is held with clinic physicians to assess their physical and psychological health state. This is now clarified in the methods (please see our response to Comment 10).

  1. In Data Analysis - Lines 104-106, were cells showed to have a frequency of 5-25? What does it imply?

To avoid issues related to small, expected cell frequencies, we used the continuity corrected Chi-square test, rather than the standard Pearsons Chi-Square test, if at least one cell had an expected frequency between 5 and 25. The reason being that when an expected cell frequency is low, the standard chi-square may not provide accurate results and in such cases, the continuity correction helps to reduce the likelihood of Type I error. We have since revised our methods of statistical analysis to improve clarity. 

OLD: The distribution of binary and nominal variables were compared using the Pearson’s Chi-square test. The Continuity corrected Chi-square test was used when one or more of the cells had an expected frequency of 5-25 and the Fisher's exact test was used when one or more of the cells had an expected frequency <5.

NEW: The associations between binary and/or nominal variables was estimated using the Pearson’s Chi-square test. However, to avoid issues related to small expected cell-frequency and to improve accuracy, we used the Continuity corrected Chi-square test if at least one cell had an expected frequency between 5 and 25, and the Fisher's exact test if at least one cell had an expected frequency <5. In the event that at least 25% of cells had an expected frequency ≤5, the Likelihood Ratio test was used.

Round 2

Reviewer 3 Report

Comments and Suggestions for Authors

I commend the authors for their prompt responsiveness to the reviewer's feedback and their conscientious efforts in revising the manuscript. The authors have effectively incorporated the requested modifications, and added new information resulting in a manuscript that now reflects the improvements suggested in the original review.